# The Effect of Guisangyou Tea on Abnormal Lipid Metabolism in Mice Induced by High-Fat Diet

**DOI:** 10.3390/foods12112171

**Published:** 2023-05-27

**Authors:** Yan Zhu, Xianghui Zhou, Nan Ling, Qiming Yu, Huijuan Wang, Qizhen Du

**Affiliations:** 1College of Food and Health Sciences, Zhejiang A & F University, Hangzhou 311300, China; yueliangyan@163.com (Y.Z.); wanghuijuan1016@126.com (H.W.); 2College of Public Health, Guilin Medical University, Guilin 541100, China; zhouxh_0472@163.com (X.Z.); lingnan0118@163.com (N.L.)

**Keywords:** Guisangyou tea, lipid metabolism, obesity, inflammation

## Abstract

This study was aimed to investigate the effect of Guisangyou tea (GSY tea) in improving abnormal lipid metabolism in mice with obesity induced by a high-fat diet (HFD). The results showed that intervention of the water extract of GSY tea (WE) decreased serum levels of lipids, and positively regulated the related antioxidant enzyme activities and the inflammatory factors in the serum and liver. In the liver, the mRNA and protein expression levels of sterol regulatory element-binding proteins-1 (SREBP-1), stearoyl-CoA desaturase-1 (SCD-1), fatty acid synthase (FASN), and acetyl CoA carboxylase (ACC) related to lipid synthesis were downregulated, and the mRNA and protein expression levels of the farnesoid X receptor (FXR) and small heterodimer partner (SHP) related to bile acid production were upregulated. The results demonstrate that GSY tea can improve abnormal lipid metabolism in obese mice by improving the body’s antioxidant capacity, regulating the inflammatory state, and reducing the synthesis of lipids and increasing the production of bile acids. GSY tea can be processed and utilized as a safe and effective resource for improving abnormal lipid metabolism.

## 1. Introduction

Obesity is a disease characterized by aberrant lipid metabolism, primarily manifested as white adipose tissue hypertrophy and excessive fat accumulation in the body [1,2]. The prevalence of overweight, obese, and severely obese individuals has increased significantly, and the age at which they develop these conditions has become younger, owing to changes in diet and lifestyle [3]. As a result, obesity has emerged as a major global health challenge [4]. Previous studies have demonstrated that the increased oxidative stress and chronic inflammatory state resulting from aberrant lipid metabolism in the liver are important pathogenic mechanisms of obesity-related metabolic syndrome. Thus, improving the body’s oxidative stress and chronic inflammatory state may facilitate obesity management [5,6]. Currently, drugs remain one of the commonly employed treatments for obesity, but most of them are synthetic drugs [7]. Synthetic drugs typically target a single entity and have limited systemic regulatory effects. Furthermore, prolonged synthetic drug use can cause several toxic side effects [8]. Consequently, the use of natural, low-toxicity, multi-pathway, multi-mode, and multi-target plant-based remedies to ameliorate aberrant lipid metabolism has garnered much attention.

Guisangyou tea (GSY tea) is made of the tender leaves of the Guisangyou mulberry tree, which is a hybrid mulberry variety bred and selected in Guangxi, China. The variety is an economically efficient mulberry with high yield and efficiency, and has a large planting area and boasts several advantages, such as wide adaptability, strong regeneration ability, high temperature resistance, drought resistance, cutting resistance, and rich nutrition [9].

Mulberry leaves are rich in numerous active ingredients beneficial to human health, such as polyphenols, flavonoids, and polysaccharides. They possess unique properties that can reduce blood lipids, exhibiting antioxidant and anti-inflammatory properties [10,11,12]. Studies have revealed that mulberry leaves can reduce blood lipids by improving the intestinal environment and accelerating the excretion of steroids and bile acids in high-fat diet (HFD) mice [13]. Mulberry leaf polyphenols can inhibit body weight and fat accumulation in mice with obesity induced by a HFD, and have potential lipid-lowering and weight-loss effects [14]. Mulberry leaves can also ameliorate lipid metabolism disorders in mice fed a long-term high-fat and high-sugar diet and possess blood-lipid-lowering, antioxidant, and anti-inflammatory properties [15].

However, there have been few reports on the effects of GSY tea on ameliorating aberrant lipid metabolism via multiple pathways and targets. This study aimed to explore the impact of GSY tea on lipid metabolism, oxidative stress, and inflammatory response in HFD-induced obese mice through in vivo animal experiments, which are important for developing the Guisangyou mulberry resource.

## 2. Materials and Methods

### 2.1. Materials and Reagents

The ordinary and high-fat diet (HFD) feeds were prepared according to the methodology outlined by YU et al. [16] and their compositions are displayed in Appendix A. Oleic acid and palmitic acid were procured from Shanghai Aladdin Biochemical Technology Co., Ltd., Shanghai, China. Free fatty acids were prepared by 2:1 ratio of oleic acid and palmitic acid by weight. The BCA protein concentration assay kit, glyceraldehyde-3-phosphate dehydrogenase (GAPDH), sterol regulatory element-binding proteins-1 (SREBP-1), stearoyl-CoA desaturase-1 (SCD-1), fatty acid synthase (FASN), acetyl CoA carboxylase (ACC), farnesoid X receptor (FXR), and small heterodimer partner (SHP) antibodies were obtained from Shanghai Biyuntian Biotechnology Co., Ltd., Shanghai, China. Total cholesterol (TC), triglyceride (TG), low-density lipoprotein cholesterol (LDL-C), high-density lipoprotein cholesterol (HDL-C), aspartate aminotransferase (AST), alanine aminotransferase (ALT), superoxide dismutase (SOD), malonic dialdehyde (MDA), catalase (CAT), glutathione peroxidase (GSH-Px), tumor necrosis factor α (TNF-α), interleukin-6 (IL-6), interleukin 1β (IL-1β), and interleukin 10 (IL-10) kits were purchased from Nanjing Jiancheng Bioengineering Institute. Trizol reagent was procured from Life Technologies Corporation. MonScriptTMRTII All-in-One Mix with dsDNase, MonAmpTMSYBR^®^Green qPCR Mix (Low ROX) kits were purchased from Mona Biotechnology Co., Ltd., Suzhou, China. Beta-actin (β-Actin), SREBP-1, SCD-1, FASN, ACC, FXR, and SHP primers were designed and synthesized by Wuhan Jinkairui Bioengineering Co., Ltd., Wuhan, China. The ECL chemiluminescence liquid was obtained from UElandy Biotechnology Co., Ltd., Suzhou, China.

### 2.2. Guisangyou Tea

GSY tea was prepared with the tender leaves of the Guisangyou mulberry tree, which were collected from Rong’an County, Liuzhou City, Guangxi in November. The leaves were washed thoroughly with clean water, and then were pan-fired for 20 min at 150–180 °C with a drum-type heater, rolled for 15 min in a rolling machine and dried at 60 °C for 12 h to yield GSY tea.

### 2.3. Experimental and Animals

Eighteen SPF-grade Kunming male mice (5 weeks old, body weight 18–22 g) were purchased from Hunan Silaikejingda Experimental Animal Co., Ltd., Changsha, China. license number SCXK (Xiang) 2019-0004.

### 2.4. Preparation of Water Extract (WE) of GSY Tea

We prepared a water extract (WE) of GSY tea for in vivo studies since GSY tea is used for a drink by brewing with hot water. The GSY tea was precisely weighed and immersed with water with a solid–liquid ratio of 1:30, and then was subsequently subjected to ultrasonication for 30 min by a SCIENTZ-950E Ultrasonic processor (Ningbo Scientz Biotechnology Co., Ltd., Ningbo, China) at 80 °C, followed by filtration to yield a water extraction solution, which was evaporated in vacuum at 55 °C, and then freeze-dried to yield WE.

### 2.5. Animal Grouping and Intervention with GSY Tea Water Extract (WE)

After adaptive feeding (1 week), mice were randomly allocated to the normal (CON) group, the high-fat diet (HFD) model group, or the WE group, with 6 mice in each group. In the conditioning period (8 weeks), the CON group received a normal diet, while the HFD and WE groups were fed a high-fat diet. Mice were housed in a breeding environment with a temperature of 22 ± 2 °C, a humidity of 50 ± 10%, and a 12 h light–dark cycle. Food and water were provided ad libitum and sterilized. After a conditioning period, in the test period (4 weeks), the WE group was administered by gavage with 0.1 mL 40 mg/kg WE (prepared with ultrapure water), while the HFD and CON groups received the same volume of water as the WE group. Body weight of mice was recorded every 2 weeks for the determination of body weight change.

### 2.6. Sample Collection

At the end of the experiment, the mice were fasted for 12 h, and their body weight was recorded. Under ether anesthesia, blood was collected from the abdominal aorta, and the organs were quickly isolated on ice, washed with PBS, and weighed accurately. Tissue samples intended for slice analysis were fixed in 10% formalin, while the remaining samples were stored in a −80 °C freezer for later use. The fat and organ indexes were calculated using the following formula:Fat/organ index/% = fat/visceral weight (g)/body weight (g) × 100

### 2.7. Determination of Serum and Liver Tissue Biochemical Indicators

Serum total cholesterol (TC), TG, high-density lipoprotein cholesterol (HDL-C), low-density lipoprotein cholesterol (LDL-C), as well as aspartate transaminase (AST), alanine transaminase (ALT), catalase (CAT), superoxide dismutase (SOD), glutathione peroxidase (GSH-Px) activity, malondialdehyde (MDA) content, tumor necrosis factor-alpha (TNF-α), interleukin-6 (IL-6), interleukin-1 beta (IL-1β), and interleukin-10 (IL-10) concentrations were determined directly according to the instructions provided in the commercial kits. For the liver tissue, after homogenization (by tissue and chilled PBS in a ratio of 1:9 (*w*/*v*)), the supernatant was obtained by centrifugation (4 °C, 8000 rpm for 15 min). The supernatant was used to determine the biochemical indicators.

### 2.8. Slice Analysis of Liver and Adipose Tissue

Adipose and liver tissues were collected, fixed in 10% formalin, and removed within 48 h. The tissues were then dehydrated using ethanol, followed by xylene treatment, and subsequently infiltrated with a series of solutions containing ethanol/xylene, xylene/paraffin, and paraffin to enable embedding. The embedded samples were sectioned into 4 μm thickness, stained with hematoxylin-eosin (HE) [17], and then examined and photographed under an optical microscope [18]. The cross-sectional area was calculated by using the image processing software Image J (image J v1.8.0, National Institutes of Health(NIH), Bethesda, USA).

### 2.9. Determination of Expression Levels of Key Genes Related to Lipid and Bile Acid Regulation in the Liver

The relative expression levels of SREBP-1, SCD-1, FASN, ACC, FXR, and SHP mRNA in liver tissue were quantified by real-time quantitative polymerase chain reaction (qRT-PCR). Total RNA was extracted from liver tissue using a reference method, and the concentration and integrity of RNA were assessed with a micro-nucleic acid protein analyzer (NanoDrop 2000). The RNA was reverse-transcribed into cDNA using the MonScriptTM RTIIII All-in-One Mix with dsDNase kit. The mRNA expression level was detected using the MonAmpTM SYBR^®^Green (Shanghai Anhe Gene Technology Co. Ltd., Shanghai, China) qPCR Mix (Low) kit and the QuanStudio 6 system. The relative gene expression was calculated using the comparative method (2^−ΔΔCt^), with Ct values normalized to β-Actin. The primer pairs for the target genes are listed in Table 1.

### 2.10. Western Blot Analysis for Determination of Protein Levels in Liver Tissue

The protein levels of key regulatory factors including SREBP-1, SCD-1, FASN, ACC, FXR, and SHP in liver tissue were determined using Western blot analysis. Briefly, liver tissues were homogenized in chilled RIPA lysis buffer (P0013C, Beyotime, Shanghai, China) containing protease inhibitors (P1046, Beyotime, Shanghai, China). The concentration of total protein was quantified using a BCA protein assay kit. Subsequently, equal amounts of protein (20 μg) were separated by electrophoresis on 10% or 12% SDS-PAGE gels and transferred onto PVDF membranes. After blocking in 5% de-fatted milk at room temperature for 2 h, the membranes were incubated with specific primary antibodies against SREBP-1, SCD-1, FASN, ACC, FXR, and SHP overnight at 4 °C. The membranes were then washed with TBST and probed with horseradish peroxidase-conjugated secondary antibodies for 1 h at room temperature. After another wash with TBST, protein bands were detected by enhanced chemiluminescence (ECL) and visualized using a chemiluminescence imaging system. GAPDH was used as an internal control. The optical density of each band was quantified using ImageJ software, and the protein expression levels were normalized to GAPDH expression.

### 2.11. Statistical Analysis

The results were expressed as mean ± standard deviation, and statistical analysis was conducted using one-way analysis of variance (ANOVA) followed by Tukey’s post hoc test in GraphPad Prism 8.0. A *p*-value of less than 0.05 was considered statistically significant.

## 3. Results

### 3.1. Effect of GSY Tea on Body Weight, Fat, and Visceral Index in Mice

As shown in Figure 1, during the conditioning period, body weight gain was significantly increased in high-fat diet mice compared with CON mice. However, there was no significant difference between HFD and WE group. During the test period, the CON and HFD mice continued to gain weight, with their growth rate gradually stabilizing, whereas the mice in the WE group exhibited a weight loss. The body weight of the WE group decreased around 10% compared with the HFD group, and returned to a similar level to that of the CON group.

The organ index provides an indirect measurement of organ damage severity [19]. As shown in Table 2 and Appendix A, compared with the CON group, the epididymal fat, perirenal fat, and liver index of HFD mice increased significantly. Compared with the HFD group, the epididymal fat, perirenal fat, and liver index in the WE group were decreased by 46.6%, 34.6%, and 11.1%, respectively, and the liver index returned to the level of the CON group. No significant differences were found in the spleen, kidney, and heart indexes among the three groups of mice (Appendix A).

### 3.2. Effect of GSY Tea on Serum and Liver Biochemical Indicators in HFD Mice

TC, TG, LDL-C, and HDL-C are commonly used clinical indicators for detecting dyslipidemia [20]. As shown in Figure 2a–d, the serum levels of TC, TG, and LDL-C were significantly increased, while the level of HDL-C was decreased in the HFD group as compared with the CON group. The serum levels of TC, TG, and LDL-C were improved to varying degrees in the WE group as compared with the HFD group. Specifically, the levels of TC and TG were reduced by 11.6% and 20.3%, respectively, and LDL-C level decreased while HDL-C level increased, although the difference was not significant. Clinically, the activities of ALT and AST in serum are sensitive indicators for detecting liver injury [21]. As shown in Figure 2e,f, the serum activities of ALT and AST were significantly increased in the HFD group as compared with the CON group. When compared with the HFD group, the serum activities of ALT and AST in the WE group were significantly decreased by 22.5% and 24.2%, respectively, and the AST activity returned to the level of CON group.

Long-term dysregulation of lipid metabolism leads to excessive accumulation of lipids in the liver, inducing a lipotoxic state that triggers the body’s oxidation pathway, producing a large amount of reactive oxygen species (ROS), and causing oxidative stress. Excessive oxidative stress is closely related to the pathogenesis of obesity-related syndromes [22]. CAT, SOD, and GSH-Px are important antioxidant enzymes in the body that play a vital role in maintaining the body’s redox balance. MDA is a sensitive indicator of free radical damage in the body and a product of lipid peroxidation [23]. As shown in Figure 3a–h, compared with the CON group, the activities of CAT, SOD, and GSH-Px in the serum and liver of mice in the HFD group were significantly decreased, while the content of MDA was significantly increased. In the WE group, the activities of CAT, SOD, and GSH-Px in the serum were increased by 20.0%, 33.8%, and 21.1%, respectively, compared with the HFD group, and returned to the levels of the CON group. Although the content of MDA showed a downward trend, there was no significant difference. The activity of CAT and SOD in the liver showed an upward trend, but also no significant difference was found. The activity of GSH-Px increased by 25.7%, while the content of MDA decreased by 22.3%. It has been reported that lipid metabolism disorders can trigger inflammation through multiple pathways, leading to an imbalance in proinflammatory and anti-inflammatory factors in the body, and an increase in inflammatory mediators [24]. In addition, TNF-α, IL-6, IL-1β, and IL-10 are important inflammatory factors in the body and are key contributors to obesity-related diseases [25], and IL-10 is a critical anti-inflammatory factor in the body that inhibits the release of various inflammatory factors and combats inflammation [26]. As depicted in Figure 4a–d, the concentrations of TNF-α, IL-6, and IL-1β in the serum of HFD group mice were increased, while the IL-10 concentration in the serum of HFD group mice was decreased, compared with the CON group. The concentrations of TNF-α, IL-6, and IL-1β in the serum of WE mice decreased by 18.4%, 16.7%, and 15.6%, respectively, compared with the HFD group, while the IL-10 concentration in the serum increased by 43.2%. All four inflammatory indicators returned to the CON group levels.

### 3.3. Effect of GSY Tea on Morphology of Hepatocyte and Epididymal Adipose Tissue

Hepatocytes in the CON group exhibited uniform size and shape, well-defined outlines, and orderly arranged nuclei (Figure 5a). Conversely, hepatocytes in the HFD group appeared enlarged, with a higher accumulation of fat vacuoles and disordered nuclei (Figure 5b). Among the WE group, the vacuoles of mice hepatocytes reduced in size, the outlines became clearer, and the nuclei gradually arranged in a more organized manner (Figure 5c). Moreover, as demonstrated in Figure 5e,f, compared with the CON group (Figure 5d), epididymal adipocytes of HFD mice displayed an extremely irregular shape, disorderly arrangement, and significantly increased cell cross-sectional area (Figure 5g). Compared with the HFD group, epididymal adipocytes of the WE group gradually regained a more regular shape, arranged in an orderly fashion, and exhibited a reduced cell cross-sectional area.

### 3.4. Effect of GSY Tea on mRNA and Protein Related to Lipid and Bile Acid

In the liver, SREBP-1, SCD-1, FASN, and ACC genes are closely related to lipid production [27], and FXR and SHP genes are closely related to bile acid synthesis [28]. The relative expression levels of SREBP-1, SCD-1, FASN, ACC mRNA, and protein in the liver of the HFD group were upregulated, while the relative expression levels of FXR, and SHP mRNA and protein in the liver of the HFD group were downregulated, compared with the CON group (Figure 6). Compared with the HFD group, SCD-1, FASN, and ACC mRNA in the liver of the WE group were downregulated by 43.3%, 41.3%, and 50.8%, respectively. FXR and SHP mRNA were upregulated by 114.3% and 54.6%, respectively. The relative mRNA expression levels of ACC, SHP, SREBP-1, FASN, and FXR recovered to the levels of the CON group. FXR and SHP protein levels were upregulated by 92.7% and 119.6% compared with the HFD group.

Correspondingly, the HFD group showed upregulated relative protein expression levels of SREBP-1, SCD-1, FASN, and ACC in mice liver, and SREBP-1, SCD-1, and FASN, and ACC protein were downregulated by 27.3%, 19.8%, 35.2%, and 16.2%, compared with the CON group (Figure 7). In contrast, the relative protein expression of FXR and SHP were downregulated in the HFD group. However, those were upregulated in the WE group by 92.7% and 119.6%, respectively. The relative protein expression levels of SCD-1, FASN, and ACC in the liver of the WE group were downregulated by 43.3%, 41.3%, and 50.8%, respectively, when compared with the HFD group. Additionally, SREBP-1, SCD-1, and FASN, ACC protein levels were downregulated by 27.3%, 19.8%, 35.2%, and 16.2% respectively, while the protein levels of FXR and SHP were upregulated by 114.3% and 54.6%, respectively, and the protein levels of FXR and SHP were upregulated by 92.7% and 119.6%, respectively. The relative protein expression levels of ACC, SHP, SREBP-1, FASN, and FXR were positively changed compared with those of the CON group.

## 4. Discussion

The effects of water extract of GSY tea (WE) on HFD-induced obese mice were studied since obesity is a complex multifactorial disorder, and long-term HFD is a major contributing factor [29]. The lard in HFD used in this experiment had a high content of 17%, and the obese mice induced by it exhibited increased body weight, fat, and liver indexes, which was consistent with previous research [30].

Obesity is a chronic, low-grade inflammatory condition that leads to long-term abnormalities in lipid metabolism, resulting in lipid accumulation in the liver [31]. This, in turn, increases the liver’s burden and causes liver damage, while also activating the body’s oxidative stress pathway and causing more severe damage to the body [32]. In the WE group, the serum levels of TC, TG, and LDL-C, and the activities of AST and ALT decreased, while the activities of antioxidant enzymes CAT, SOD, and GSH-Px increased compared with the obesity model group. Moreover, the concentration of pro-inflammatory factors TNF-α, IL-6, and IL-1β decreased, and the concentration of anti-inflammatory factor IL-10 increased. These results suggest that WE can improve abnormal lipid metabolism in HFD mice by enhancing the body’s antioxidant capacity and regulating the inflammatory state, which are consistent with previous studies that have shown the effectiveness of mulberry leaf extracts in improving blood lipid disorders in rats with atherosclerosis [33] and in attenuating clinical signs, reducing inflammatory cytokine secretion, and inhibiting inflammatory pathway activation in mice with colitis induced by dextran sulfate sodium [34].

The liver plays a crucial role in lipid and bile acid synthesis in humans. Liver damage can lead to impaired utilization and metabolism of lipids and bile acids, resulting in abnormal lipid metabolism in the body. In the hepatic levels of the WE group, the protein expression of SREBP-1 was downregulated, along with downregulation of the mRNA expression and protein levels of SCD-1, FASN, and ACC, and the mRNA and protein expression levels of FXR and SHP in the liver were upregulated. It has been reported that SREBP-1, SCD-1, and FASN are involved in the regulation of lipid synthesis, and their abnormal expression can lead to lipid metabolism disorders [28]. ACC catalyzes the production of malonyl-CoA, which is involved in the synthesis of fatty acids [35]. Upon FXR activation in the liver, downstream SHP is also involved in regulating bile acid synthesis [36]. Our results involving ACC, FXR, and SHP support that WE possesses a positive effect in improving abnormal lipid metabolism.

## 5. Conclusions

The water extract of Guisangyou tea (WE) effectively lowers serum lipid levels, regulates enzyme activities, and improves abnormal lipid metabolism in obese mice. It decreases the expression of key genes (SREBP-1, SCD-1, FASN, ACC) involved in lipid synthesis, while enhancing the expression of FXR and SHP proteins associated with bile acid production. These findings highlight the significant impact of GSY tea on rectifying abnormal lipid metabolism in obese mice. Therefore, GSY tea can be processed and utilized as a safe and effective resource for improving abnormal lipid metabolism.

## Figures and Tables

**Figure 1 foods-12-02171-f001:**
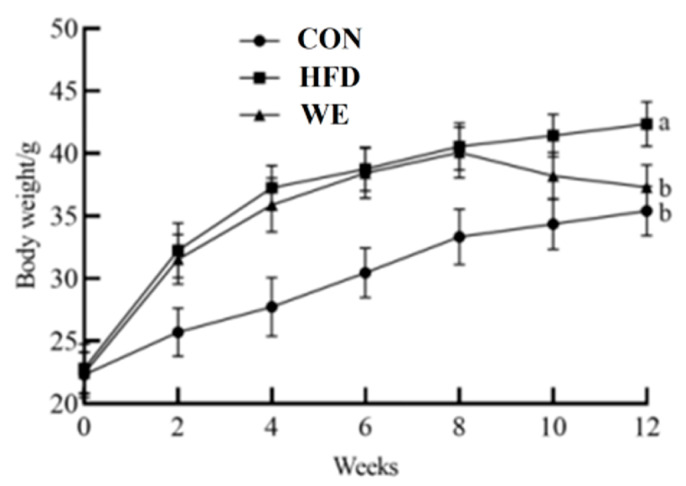
The mice body weight of the control group (CON), high-fat diet group (HFD), and high-fat diet-WE treatment group (WE). Significant differences between groups are denoted by different letters (*p* < 0.05).

**Figure 2 foods-12-02171-f002:**
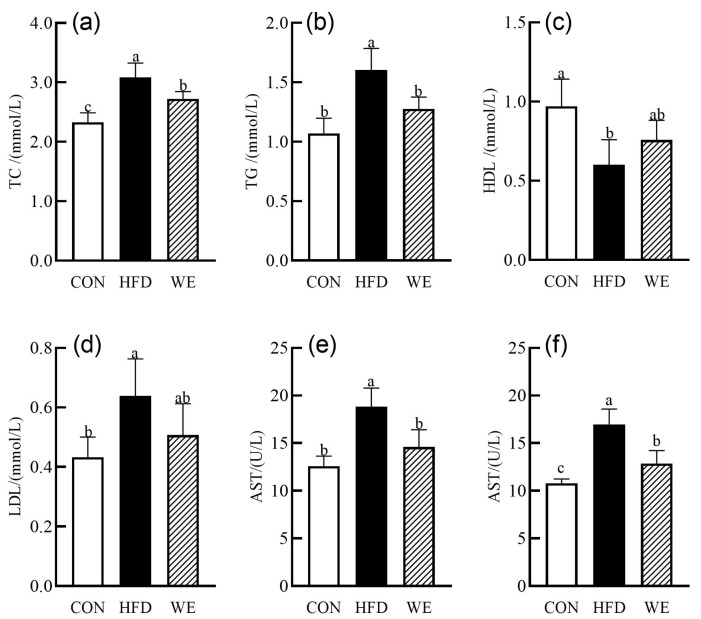
The concentration of lipids and enzyme activity in mice serum of the control group (CON), high-fat diet group (HFD), and high-fat diet-WE treatment group (WE). (**a**) TC, (**b**) TG, (**c**) HDL-C, (**d**) LDL-C, (**e**) AST, (**f**) ALT. Significant differences between groups are denoted by different letters (*p* < 0.05).

**Figure 3 foods-12-02171-f003:**
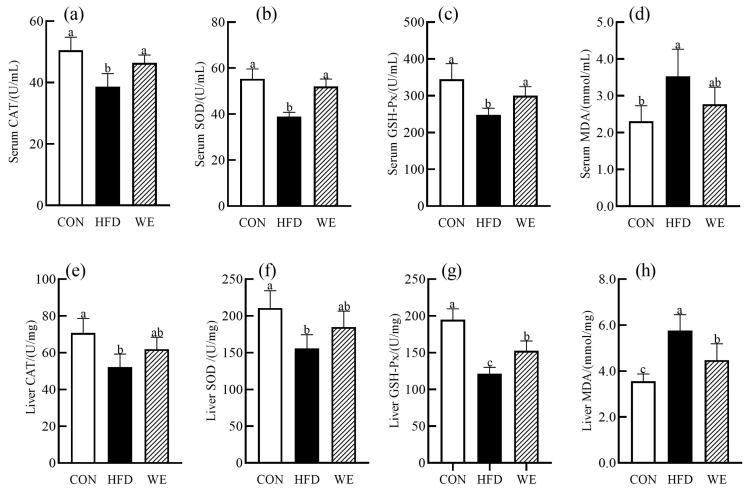
The activities of the oxidation biochemical indexes in the mice of the control group (CON), high-fat diet group (HFD), and high-fat diet-WE treatment group (WE). (**a**) CAT in serum, (**b**) SOD in serum, (**c**) GSH-Px in serum, (**d**) MAD in serum, (**e**) CAT in liver, (**f**) SOD in liver, (**g**) GSH-Px in liver, (**h**) MAD in liver. Significant differences between groups are denoted by different letters (*p* < 0.05).

**Figure 4 foods-12-02171-f004:**
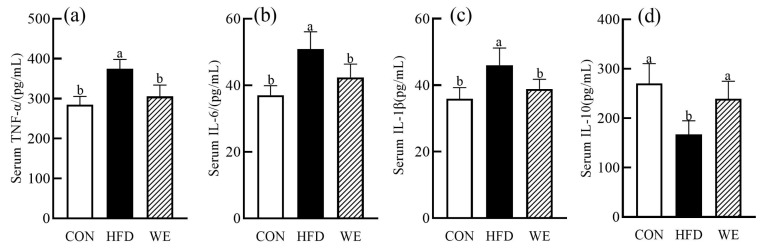
The level of the inflammatory biochemical indexes in mice serum of the control group (CON), high-fat diet group (HFD), and high-fat diet-WE treatment group (WE). (**a**) TNF-α, (**b**) IL-6, (**c**) IL-1β, (**d**) IL-10. Significant differences between groups are denoted by different letters (*p* < 0.05).

**Figure 5 foods-12-02171-f005:**
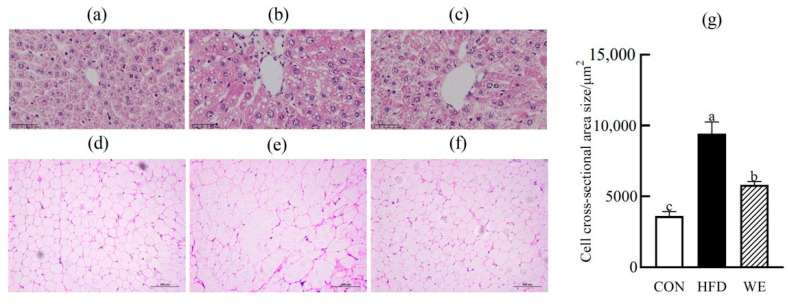
The morphology of liver and epididymal adipose tissue in the mice of the control group (CON), high-fat diet group (HFD), and high-fat diet-WE treatment group (WE). (**a**) Control, liver tissue; (**b**) HFD, liver tissue; (**c**) WE, liver tissue; (**d**) control, epididymal fat; (**e**) HFD, epididymal fat tissue; (**f**) WE, epididymal fat tissue; (**g**) cell cross-sectional area size/μm^2^. Significant differences between groups are denoted by different letters (*p* < 0.05).

**Figure 6 foods-12-02171-f006:**
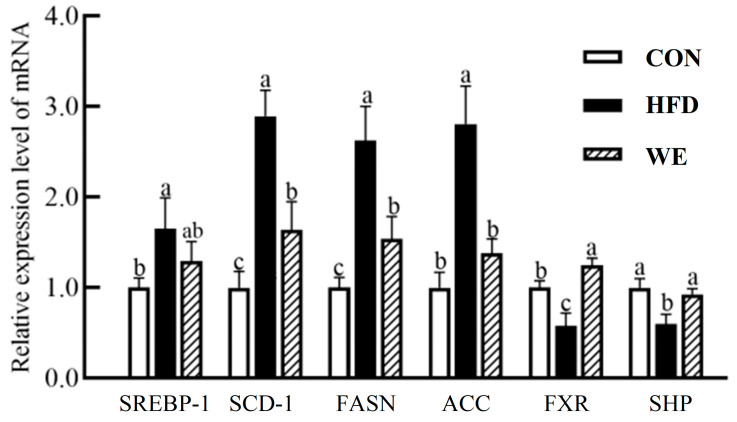
The relative expression of the gene mRNAs related to lipid and bile acid regulation in the mice of the control group (CON), high-fat diet group (HFD), and high-fat diet-WE treatment group (WE). Significant differences between groups are denoted by different letters (*p* < 0.05).

**Figure 7 foods-12-02171-f007:**
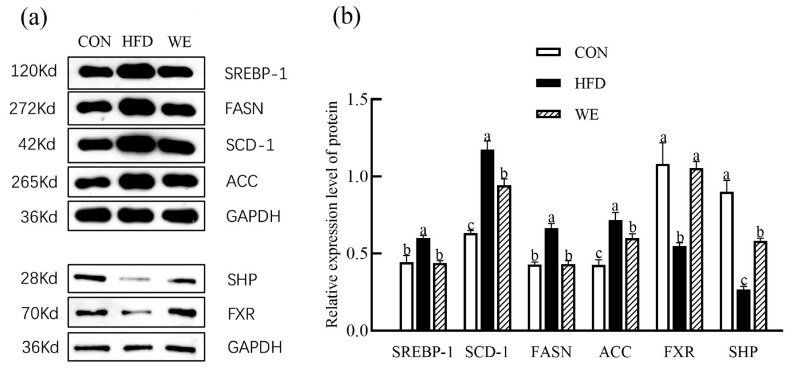
The relative expression of proteins related to lipid and bile acid regulation in mice livers of the control group (CON), high-fat diet group (HFD), and high-fat diet-WE treatment group (WE). (**a**) Protein level by Western blot. (**b**) Relative levels normalized to GAPDH. Significant differences between groups are denoted by different letters (*p* < 0.05).

**Table 1 foods-12-02171-t001:** Primer sequences of the target genes (mouse source).

Gene Number	Forward Primer (5′-3′)	Reverse Primer (5′-3′)
β-Actin	TCATCACTATTGGCAACGAGC	AACAGTCCGCCTAGAAGCAC
SREBP-1	CACCCTGTAGGTCACCGTTT	GCTCGCTCTAGGAGATGTTCA
SCD-1	GCGATACACTCTGGTGCTCA	TGGTAGTTGTGGAAGCCCTC
FASN	CTGGCATTCGTGATGGAGTC	GGGCAGAAGGTCTTGGAGAT
ACC	TTGGCCCTGTTGAGCATCTTT	GCCCTCTTTGTACCAGTGACG
FXR	CGAGATGCCTGTGACAAAGA	GCAGACCACACACAGCTCAT
SHP	TGTCCTAGCCAAGACAGTAGCC	ACCTCGAAGGTCACAGCATC

**Table 2 foods-12-02171-t002:** The mouse fat indexes of the control group (CON), high-fat diet group (HFD), and high-fat diet-WE treatment group (WE).

Group	Fat Index/%	
	Epididymal Fat	Perirenal Fat
CON	0.956 ± 0.195 ^c^	0.148 ± 0.057 ^c^
HFD	3.790 ± 0.306 ^a^	0.416 ± 0.032 ^a^
WE	2.022 ± 0.464 ^b^	0.272 ± 0.045 ^b^

Significant differences between groups are denoted by different letters (*p* < 0.05).

## Data Availability

Data are contained within the article or Appendix A.

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
