# Peer review of "The Effect of Guisangyou Tea on Abnormal Lipid Metabolism in Mice Induced by High-Fat Diet"

_foods, 2023, doi:10.3390/foods12112171_

Round 1

Reviewer 1 Report

Review: Foods2389384

Comments and Suggestions for Authors (will be shown to authors)

     This manuscript describes the study to characterize the protective effects of Guisangyou tea against the bad influence of high fat diet. The results would be important to evaluate the detail of the benefits of the tea. Some comments are following:

• Abstract

Line 20: The statement “by improving the body’s antioxidant capacity” is not support here because the previous portion of the Abstract does not include issues related to in body’s antioxidant capacity.

• Materials and Methods

Line 68, “Free fatty acids (FFA) were prepared from Oleic acid and Palmitic acid at a ratio of 2:1.”: Were FFAs purified from a (commercial) mixture? 

Line 95: Subsection of “2.4. Experimental method” is not necessary. You can continue just “2.4. Preparation of water extract (WE) of GSY tea,” “2.5. Animal grouping and intervention with GSY tea water extract (WE),” and so on.

Line 104, 2.4.2. Animal grouping and intervention with GSY tea water extract (WE):

I strongly recommend putting names to the three feeding periods: for example, adaptive period (1 week), conditioning period (8 weeks) of 3 groups, and test period (4 weeks, with or without WE). Using the period names, you can easily indicate the period you’re talking about when you explain the experiments and results of each period separately.

Line 110: The way of feeding of WE is unclear. Was it administrated directly into stomach, or ingested ad libitum in water or in diet? 

Line 130-144, 2.3.4.: The explanations are complicated because similar explanations appear separately. How about separating sample preparation and measurement portions. 

Line 133: The detailed method or condition of preparing cell-free liver extracts (i.e., homogenization and centrifugation) should be explained.

Line 171: “RIPA lysis buffer” and “protease inhibitors” should be explained in detail.

• Redundant paragraphs: There are many cases of two redundant successive paragraphs with almost identical contents. The second paragraphs often seem to be improved versions of the first paragraphs. 

   2.4.2., 2.3.5., 2.3.7., 

• Results

Line 205: What are “trans-life behaviors”? Detailed explanation is required.

Line 214: The calculation giving “11.9%” is unclear.

Line 217: Should be “As shown in Table 2 and Table 2S” because the following sentence mention results from both Tables.

Line 274, Figure 2: Labels of “Concentration of serum” and “Activity of serum” in the vertical axes are not necessary. Figure legend and units well explain the meanings.

Line 280, Figure 3: Labels of “activity of” and “Content of” in the vertical axes are not necessary. Figure legend and units well explain the meanings.

Line 293-297: The statement cannot be supported by the data, because the magnification of Figure 5a, 5b and 5c is not large enough to show the difference in the size and shape of hepatocytes between the sample groups. Therefore, Figure 5a, 5b and 5c should be replaced with those of higher magnification.

Line 299: There is no explanation how the cell cross-sectional area was determined.

Line 316-332: The explanation is confusing Figure 6 and probably Figure 7 data.

Line 327: “Figure 7” should be added here.

Line 331: “mRNA” should be deleted.

Line 330-332: The explanation is for mRNA levels, not protein, which was already done in the previous paragraph for Figure 6, not Figure 7.

Line 338, Figure 7(a): What is “MLW”? Does “MLW” mean “WE”?

• Figures

In bar graphs, labels of vertical axes are unnecessarily long. “Concentration of” and “Activity of” are not necessary. Figure legends and units well explain the meanings of the axes.

• Discussion: Any expected mechanism leading to the present results, as well as candidates of the active components in the Guisangyou tea and their biochemical activity, would be a great help to develop usage of the tea.

• References: Some references do not seem to support the issues for which they are citated.

Line 29, Reference 1: This reference does not seem to be suitable here as the first reference expected to describe general issues of obesity in this manuscript, because its abstract available in PubMed indicates that this is an original paper demonstrating effects of dietary curcumin on obese model mice. Probably, we could not expect enough general information about obesity. (I said “probably,” because I cannot look at the full text of this reference in my workplace.)

Line 161, Reference 18: This reference 8 is not suitable here because it does not describe the method of total RNA extraction, which the present manuscript requires here.

Line 165, Reference 19: This reference is not suitable here because it does not describe relative gene expression.

Line 216, Reference 20: This reference is not suitable here because the reference describes liver function indices but not organ index discussed in the present manuscript.

Line 312: Reference 28 is not suitable here because it does not include SCD-1.

Line 316-332: The explanation is confusing Figure 6 and probably Figure 7 data.

Line 346, Reference 30: This reference is not suitable here because it does not mainly describe that “obesity is a complex multifactorial disorder, and long-term HFD is a major contributing factor.” 

Line 365, Reference 36: This reference is not suitable here, because the fact that “The liver plays a crucial role in lipid and bile acid synthesis in humans” have been established before publication of reference 36.

Line 373: Reference 37 is not suitable here because the reference does not describe SREBP-1, SCD-1 and SHP.

Line 374, Reference 38: This reference is not suitable here because the reference does not directly describe the established classical fact stated in this sentence.

Line 375, Reference 39: This reference is not suitable here because the reference does not directly describe the relationship between FXR, SHP and bile caid synthesis.

Review: Foods2389384

Comments on the Quality of English Language (will be shown to authors)

     The quality of English language is not at the submission level. 

Abstract

Line 21: The statement “reducing the synthesis of … bile acids” contradict the previous statement “the mRNA and protein expression levels … were up-regulated (line 18-19).”

2. Materials and Methods

Line 108: “Breeding environment was …, … and ….”    

Line 112: “and the body weight…”.

Line 128: Determine which is used in allover the manuscript, “indices” or “indexes.” Indices: Line128; Indexes: line 228, line 56 (Table 2S),line 348, and so on.

Line 136-142: Grammatically confusing.

3. Results

Line 203: The word “the drug intervention” is bad here, because the meaning of “drug” here is quite unclear. The sentence at the line 203 will apparently mislead the readers to recognize that the CON group mice were treated with “the drug.” But this is not true. 

Line 206 and others: Connecting with “After WE intervention” or “Following WE intervention” is logically wrong. These words suggest the comparison is between before and after WE intervention. But the actual comparison is with and without WE intervention. In most cases, the words “After WE intervention” or “Following WE intervention” should be replaced with “In the WE group.”

Line 228: “organ indexes” should be “fat indexes.”

4. Discussion

Line 348: The sentence “Furthermore, …” does not seem to logically follow the previous sentence, because the previous sentence only describes the effect of lard, but not the effects of WE.

Line 356: “the obesity model group” is helpful.

5. Conclusion

Line 382: The meaning of “and the mRNA” is not clear.

Line 378-385: This sentence is too long and complicated. 

Author Response

 This manuscript describes the study to characterize the protective effects of Guisangyou tea against the bad influence of high fat diet. The results would be important to evaluate the detail of the benefits of the tea. Some comments are following:

Abstract

Line 20: The statement “by improving the body’s antioxidant capacity” is not support here because the previous portion of the Abstract does not include issues related to in body’s antioxidant capacity.

Thank you for the suggestion provided by the reviewer. In the abstract, we mentioned "enzyme activities" without specifically highlighting antioxidant enzymes, which may lead to a misunderstanding for readers. We have made the necessary modification to "antioxidant enzyme activities" to ensure coherence with the conclusion that follows, which states "by improving the body's antioxidant capacity."

Materials and Methods

Line 68, “Free fatty acids (FFA) were prepared from Oleic acid and Palmitic acid at a ratio of 2:1.”: Were FFAs purified from a (commercial) mixture? 

Thank you for pointing out this concern. We mentioned the sources of Oleic acid and Palmitic acid earlier. We have modified the descriptive statement to "Free fatty acids were prepared by a weight ratio of 2:1 of oleic acid and palmitic acid."

Line 95: Subsection of “2.4. Experimental method” is not necessary. You can continue just “2.4. Preparation of water extract (WE) of GSY tea,” “2.5. Animal grouping and intervention with GSY tea water extract (WE),” and so on.

The reviewer provided valuable feedback, and we have followed their suggestion by removing the section titled "2.4. Experimental method." Furthermore, we have made the necessary modifications to the subsequent content as per their advice.

Line 104, 2.4.2. Animal grouping and intervention with GSY tea water extract (WE):

I strongly recommend putting names to the three feeding periods: for example, adaptive period (1 week), conditioning period (8 weeks) of 3 groups, and test period (4 weeks, with or without WE). Using the period names, you can easily indicate the period you’re talking about when you explain the experiments and results of each period separately.

Line 110: The way of feeding of WE is unclear. Was it administrated directly into stomach, or ingested ad libitum in water or in diet? 

Based on the excellent feedback provided by the reviewer, we have made the following improvements. Firstly, we have introduced distinct names for each treatment period and included additional explanations regarding the operational cycles. This will facilitate a more accurate description of the treatment process during the presentation and discussion of the results in the later stages. Furthermore, we realized that we had not clarified the method of administering the extract treatment. Therefore, we have added a description of oral gavage administration in the Methods section.

Line 130-144, 2.3.4.: The explanations are complicated because similar explanations appear separately. How about separating sample preparation and measurement portions. 

Line 133: The detailed method or condition of preparing cell-free liver extracts (i.e., homogenization and centrifugation) should be explained.

Line 171: “RIPA lysis buffer” and “protease inhibitors” should be explained in detail.

Redundant paragraphs: There are many cases of two redundant successive paragraphs with almost identical contents. The second paragraphs often seem to be improved versions of the first paragraphs. 

   2.4.2., 2.3.5., 2.3.7., 

We appreciate the valuable suggestions provided by the reviewer. We have recognized the oversight that led to repeated paragraphs during the revision process, resulting in poor readability and lack of clarity in several sections of the Methods. To address this issue, we have removed the duplicated paragraphs that arose from the modifications made during the paper revision. Additionally, we have followed the reviewer's modification suggestions and added supplementary explanations regarding the omitted steps and reagent information in the sample processing procedures.

Results

Line 205: What are “trans-life behaviors”? Detailed explanation is required.

Line 214: The calculation giving “11.9%” is unclear.

We have thoroughly examined the section in question within the results. We have acknowledged the issues pointed out by the reviewer as well as identified additional concerns. In response, we have carefully revised the animal experimentation process, ensuring that it aligns with the logical flow of the experiment and addressing any shortcomings.

Line 217: Should be “As shown in Table 2 and Table 2S” because the following sentence mention results from both Tables.

Line 274, Figure 2: Labels of “Concentration of serum” and “Activity of serum” in the vertical axes are not necessary. Figure legend and units well explain the meanings.

Line 280, Figure 3: Labels of “activity of” and “Content of” in the vertical axes are not necessary. Figure legend and units well explain the meanings.

We appreciate the reviewer for pointing out the issues, and we have made the necessary corrections accordingly.

Line 293-297: The statement cannot be supported by the data, because the magnification of Figure 5a, 5b and 5c is not large enough to show the difference in the size and shape of hepatocytes between the sample groups. Therefore, Figure 5a, 5b and 5c should be replaced with those of higher magnification.

Line 299: There is no explanation how the cell cross-sectional area was determined.

We would like to express our gratitude to the reviewer for highlighting the issue. As per their suggestion, we have included relevant explanations in the Methods section.

Line 316-332: The explanation is confusing Figure 6 and probably Figure 7 data.

Line 327: “Figure 7” should be added here.

Line 331: “mRNA” should be deleted.

Line 330-332: The explanation is for mRNA levels, not protein, which was already done in the previous paragraph for Figure 6, not Figure 7.

Line 338, Figure 7(a): What is “MLW”? Does “MLW” mean “WE”?

The description in this section was somewhat confusing, and we appreciate the reviewer for raising this concern. During the revision process, we have restructured the content to provide separate descriptions for mRNA and protein levels, thus avoiding any confusion. Additionally, we have made the necessary modification in the figure, replacing "MLW" with "WE" throughout.

Figures

In bar graphs, labels of vertical axes are unnecessarily long. “Concentration of” and “Activity of” are not necessary. Figure legends and units well explain the meanings of the axes.

We appreciate the reviewer for pointing out the issue. In response, we have made appropriate adjustments to the tables in the manuscript.

Discussion: Any expected mechanism leading to the present results, as well as candidates of the active components in the Guisangyou tea and their biochemical activity, would be a great help to develop usage of the tea.

Thank you for the questions posed by the reviewers. Mulberry leaves are rich in various polyphenolic compounds, namely chlorogenic acid, rutin, isoquercitrin, quercetin, astragalin, and kaempferol. Furthermore, mulberry leaves also contain other flavonoids and polyphenols, such as deoxynojirimycin and mulberry anthocyanins.

References: Some references do not seem to support the issues for which they are citated.

Line 29, Reference 1: This reference does not seem to be suitable here as the first reference expected to describe general issues of obesity in this manuscript, because its abstract available in PubMed indicates that this is an original paper demonstrating effects of dietary curcumin on obese model mice. Probably, we could not expect enough general information about obesity. (I said “probably,” because I cannot look at the full text of this reference in my workplace.)

We appreciate the feedback from the reviewer. In response to their suggestion, we have revised the reference to a review article that encompasses studies related to obesity in the population, making it more appropriate for the context.

Line 161, Reference 18: This reference 8 is not suitable here because it does not describe the method of total RNA extraction, which the present manuscript requires here.

Line 165, Reference 19: This reference is not suitable here because it does not describe relative gene expression.

Line 216, Reference 20: This reference is not suitable here because the reference describes liver function indices but not organ index discussed in the present manuscript.

We appreciate the reviewer's feedback. The RNA extraction method used in our experiment is indeed a commonly employed method. Therefore, we have removed two references accordingly.

Line 312: Reference 28 is not suitable here because it does not include SCD-1.

We have found a more relevant literature source that better supports the content in this section. Consequently, we have replaced the previous reference (No. 28) with the reference from Xiao et al..

Line 346, Reference 30: This reference is not suitable here because it does not mainly describe that “obesity is a complex multifactorial disorder, and long-term HFD is a major contributing factor.” 

We have conducted further research and found a more informative literature source. As a result, we have replaced the previous reference (No. 30) with the reference from Rodriguez et al.

Line 365, Reference 36: This reference is not suitable here, because the fact that “The liver plays a crucial role in lipid and bile acid synthesis in humans” have been established before publication of reference 36.

We appreciate the reviewer's suggestion. As this is widely accepted knowledge, we have removed the reference at this particular section.

Line 373: Reference 37 is not suitable here because the reference does not describe SREBP-1, SCD-1 and SHP.

We have replaced reference 37 with reference 28, as suggested by the reviewer. As described by the reviewer, the new reference provides a more comprehensive source to support our claims.

Line 374, Reference 38: This reference is not suitable here because the reference does not directly describe the established classical fact stated in this sentence.

Line 375, Reference 39: This reference is not suitable here because the reference does not directly describe the relationship between FXR, SHP and bile caid synthesis.

We appreciate the reviewer's input. In response, we have conducted a thorough literature search and carefully considered the need for references in these two instances. We have replaced them with more relevant references that have a high degree of correlation to the topic at hand.

 The quality of English language is not at the submission level. 

We have carefully checked the grammar and improved our writing skills to meet the requirements of the journal.

Reviewer 2 Report

The study investigated the effect of tea extract on improvement in lipid homeostasis in mouse model of obesity. The manuscript is OK but need major revisions.

Line-102- theno??

Line-108- sentence incomplete 

Line- table 1- CPT1 in primer list but not in results anywhere. 

Line - 141- how was the mental state determined 

Line-197-199- Statistical analysis should be indicated be indicated by *. Please change the ranks and add brackets to indicate differences between groups.

Line -209- Do you mean before intervention?

Line - 217-218- liver weight not given in table-2 although mentioned in the text. Please modify sentence.

Line-327-337- repetition of what has been already said in 316-322. Please remove repetitions. 

What is the reason for the weight loss?

Where is the food intake data?

Was brown fat collected and analyzed for any thermogenic genes?

Most conventional studies have not been able to show increased plasma levels of LDL cholesterol with high fat diet. However this study reported that the LDL cholesterol levels increase with HFD in their mouse model. How many cohorts the study was performed on?

Author Response

1. Abstract

Line 21: The statement “reducing the synthesis of … bile acids” contradict the previous statement “the mRNA and protein expression levels … were up-regulated (line 18-19).”

 We greatly appreciate the reviewer for pointing out the error in this section. After FXR activation, there is an increase in cholesterol utilization and bile acid synthesis. We have made the necessary revisions accordingly.

2. Materials and Methods

Line 108: “Breeding environment was …, … and ….”    

Line 112: “and the body weight…”.

Line 128: Determine which is used in allover the manuscript, “indices” or “indexes.” Indices: Line128; Indexes: line 228, line 56 (Table 2S),line 348, and so on.

Line 136-142: Grammatically confusing.

 We appreciate the valuable suggestions provided by the reviewer. We have acknowledged the oversight that led to repeated paragraphs during the revision process, resulting in poor readability and lack of clarity in several sections of the Methods. We have removed the duplicated paragraphs that arose from the modifications made during the paper revision. Additionally, we have ensured consistency by using "indexes" throughout the manuscript.

3. Results

Line 203: The word “the drug intervention” is bad here, because the meaning of “drug” here is quite unclear. The sentence at the line 203 will apparently mislead the readers to recognize that the CON group mice were treated with “the drug.” But this is not true. 

Line 206 and others: Connecting with “After WE intervention” or “Following WE intervention” is logically wrong. These words suggest the comparison is between before and after WE intervention. But the actual comparison is with and without WE intervention. In most cases, the words “After WE intervention” or “Following WE intervention” should be replaced with “In the WE group.”

Line 228: “organ indexes” should be “fat indexes.”

We greatly appreciate the meticulous review by the reviewer, who pointed out several errors. We have thoroughly examined the manuscript and made the necessary corrections regarding expressions such as "After WE intervention" or "Following WE intervention" throughout the entire paper.

4. Discussion

Line 348: The sentence “Furthermore, …” does not seem to logically follow the previous sentence, because the previous sentence only describes the effect of lard, but not the effects of WE.

Line 356: “the obesity model group” is helpful.

We sincerely appreciate the suggestions provided by the reviewer. In response to these two recommendations, we have made the necessary changes.

5. Conclusion

Line 382: The meaning of “and the mRNA” is not clear.

Line 378-385: This sentence is too long and complicated. 

We appreciate the reviewer for pointing out the issue in this section. We have carefully examined this part and made the necessary modifications to eliminate any ambiguity.

The study investigated the effect of tea extract on improvement in lipid homeostasis in mouse model of obesity. The manuscript is OK but need major revisions.

Line-102- theno??

Line-108- sentence incomplete 

Line- table 1- CPT1 in primer list but not in results anywhere. 

We appreciate the reviewer for highlighting the issues. We have addressed the problems mentioned in these sections. Regarding line 108, the repetition was unintentional and resulted from the previous modifications made to the paper. Therefore, we have retained the polished paragraph. As for CPT-1, we only examined it at the RNA level and did not investigate the protein level, which caused inconsistency. Therefore, we have removed this information from the table as well as from the final manuscript.

Line - 141- how was the mental state determined 

We appreciate the reviewer for pointing out the issue. We have removed the description of the animals' mental state in this section. During the animal experiment, we conducted daily checks on the animals' condition. However, since these were routine checks performed on a daily basis, there is no need to mention them explicitly in the manuscript.

Line-197-199- Statistical analysis should be indicated be indicated by *. Please change the ranks and add brackets to indicate differences between groups.

Line -209- Do you mean before intervention?

We have followed the advice of another reviewer and rephrased the description of different treatment time points for the animals in this section to avoid confusion.

Line - 217-218- liver weight not given in table-2 although mentioned in the text. Please modify sentence.

Line-327-337- repetition of what has been already said in 316-322. Please remove repetitions. 

We sincerely appreciate the meticulous review by the reviewer. It is true that in our initial draft, we used similar expressions, which led to the perception of duplication in these two instances where mRNA and protein levels were supposed to be different. Another reviewer also pointed out the issue with our wording in these two sections, and we have thoroughly examined and made the necessary corrections.

What is the reason for the weight loss?

Where is the food intake data?

In this case, we recorded feed intake data during the experimental process. There were differences in feed intake between the CON and HFD groups, with CON group showing higher intake. However, there were no significant differences in feed intake observed in mice fed a high-fat diet. Therefore, we did not include this information in the manuscript.

Was brown fat collected and analyzed for any thermogenic genes?

Thank you for the valuable suggestion provided by the reviewer. Indeed, in this particular study, we did not include the measurement of thermogenic genes in brown fat. However, we have taken note of this limitation and have conducted the necessary measurements in subsequent experiments. In our future studies related to this topic, we will include data on thermogenic genes.

Most conventional studies have not been able to show increased plasma levels of LDL cholesterol with high fat diet. However this study reported that the LDL cholesterol levels increase with HFD in their mouse model. How many cohorts the study was performed on?

Thank you for your attention to our study. Indeed, our animal model did exhibit significant differences in body weight during the high-fat feeding period. Similar results have been reported in many related studies, where LDL-c levels were found to be significantly increased in mice fed a high-fat diet. As an example, I would like to refer to the following article: [https://www.ncbi.nlm.nih.gov/pmc/articles/PMC7580858/pdf].

Round 2

Reviewer 2 Report

The author has addressed the suggestions.